# Immunogenicity of COVID-eVax Delivered by Electroporation Is Moderately Impacted by Temperature and Molecular Isoforms

**DOI:** 10.3390/vaccines11030678

**Published:** 2023-03-16

**Authors:** Federico D’Alessio, Lucia Lione, Erika Salvatori, Federica Bucci, Alessia Muzi, Giuseppe Roscilli, Mirco Compagnone, Eleonora Pinto, Gianfranco Battistuzzi, Antonella Conforti, Luigi Aurisicchio, Fabio Palombo

**Affiliations:** 1Takis, 00128 Rome, Italy; 2Neomatrix, 00128 Rome, Italy; 3Alfasigma, 30400 Pomezia, Italy; 4Evvivax, 00128 Rome, Italy

**Keywords:** COVID-19, DNA stability, immune responses, vaccines

## Abstract

DNA integrity is a key issue in gene therapy and genetic vaccine approaches based on plasmid DNA. In contrast to messenger RNA that requires a controlled cold chain for efficacy, DNA molecules are considered to be more stable. In this study, we challenged this concept by characterizing the immunological response induced by a plasmid DNA vaccine delivered using electroporation. As a model, we used COVID-eVax, a plasmid DNA-based vaccine that targets the receptor binding domain (RBD) of the SARS-CoV-2 spike protein. Increased nicked DNA was produced by using either an accelerated stability protocol or a lyophilization protocol. Surprisingly, the immune response induced in vivo was only minimally affected by the percentage of open circular DNA. This result suggests that plasmid DNA vaccines, such as COVID-eVax that have recently completed a phase I clinical trial, retain their efficacy upon storage at higher temperatures, and this feature may facilitate their use in low-/middle-income countries.

## 1. Introduction

Non-viral vaccines (such as DNA and RNA) against severe acute respiratory syndrome coronavirus 2 (SARS-CoV-2) reached global diffusion with the COVID-19 pandemic, and their significant success in preventing infection and deaths was achieved in a fast unprecedented timeframe due to favorable characteristics such as rapid turnaround production, low toxicity, and high immunogenicity [1]. Their rapid design, construction, and manufacturing make this class of vaccines an ideal vaccination platform for pandemic preparedness [2]. To this aim, two main requirements must be fulfilled: high purity and stability of the final formulation, both achievable through proper manufacturing and storage conditions. Plasmid DNA as a drug must be highly purified to remove all residual bacterial proteins and the host DNA, and, for extended stability at a convenient storage temperature, contaminating nucleases must be removed [3]. Among its distinctive advantages, plasmid DNA readily and quickly renatures under many conditions with no loss of biological activity whereas it usually requires chemical modification to produce an irreversible loss of biological activity, thus indicating DNA chemical integrity as the primary endpoint of structural stability studies. In principle, this offers the opportunity to replace measurements of biological activity with high-resolution chemical analysis and, therefore, to evaluate plasmid DNA-based products similarly to small molecule drugs rather than biological entities.

Monitoring the stability of plasmid DNA over long periods is mainly dependent on the chemical integrity of the phosphodiester backbone. The introduction of a single break in the DNA backbone converts supercoiled (SC) plasmid DNA to open circular (OC) plasmid DNA, or a double nick on both strands leads to linear (LN) DNA. Monitoring the conversion of SC to OC DNA provides a convenient and sensitive assay to assess DNA stability. There are different methods to monitor the conversion of SC plasmid DNA to the OC and LN forms of DNA, such as agarose gel electrophoresis and various chromatographic techniques. These two methods are included in the release tests of injectable plasmid DNA [4,5].

DNA delivery methods are key to achieving optimal gene expression and immunogenicity. Electroporation (EP) is an effective method that, by permeabilizing cells, allows the DNA to reach the nucleus and express the target antigen [6]. In vivo EP requires optimization of the electrical conditions according to the target tissue, with most of the vaccine applications carried out in the muscle. DNA delivery by EP (DNA-EP) has been used for many different applications including viral vaccines [7,8], cancer vaccines targeting either tumor-associated antigens [9,10] or neoantigens [11,12], or gene therapy for the delivery of plasmid DNA encoding therapeutic monoclonal antibodies [13,14]. The advantageous use of plasmid DNA, in comparison to conventional vaccination platforms, namely inactivated, attenuated, or subunit vaccines, is justified by the short turnaround time from the identification of the target to the delivery in the muscle and by the limited cost of production. Other delivery systems based on viral backbones require the use of mammalian cells with higher costs and the potential interference of the expressed transgene with defective viral growth [15]. Moreover, one recent evolution of DNA-EP technology is the use of complete synthetic DNA, which does not require the use of bacterial growth [16,17,18]. However, these technologies are still at the preclinical stage likely due to the lack of a large-scale production process as instead available for plasmid DNA production.

Several DNA-based vaccines have been so far licensed for veterinary applications [19], and during the SARS-CoV-2 pandemic, a plasmid DNA-based vaccine, ZyCoV-D, was authorized for emergency use (EUA) by the office of the Drug Controller General of India (DCGI) for preventing COVID-19 in humans [20]. Very recently, we have completed a phase I clinical trial with COVID-eVax, a naked plasmid DNA expressing the receptor binding domain (RBD) of the SARS-CoV-2 spike protein and delivered by EP, first shown to elicit potent immune responses and protection in animal models [21,22,23] and then proven to be safe and immunogenic also in humans [24].

Although one of the well-noted advantages of DNA is its stability, which is a function of the buffer composition and storage temperatures [25], nicked DNA increases as a function of the temperature, requiring a cold chain to maintain DNA integrity, and according to Pharmacopeia, SC plasmid DNA in injectable DNA must be higher than 80% [3]. However, the impact of nicked DNA on biological functions such as gene expression or in vivo immune response has not been evaluated in the context of electroporation delivery.

Therefore, in this study, we characterize the impact of either the nicked DNA produced in an accelerated stability test [26] or lyophilization on COVID-eVax expression levels by performing an in vitro potency assay and evaluating the immune response induced in a mouse model upon intramuscular EP.

## 2. Materials and Methods

### 2.1. Plasmid DNA Manufacturing

The production of the engineering run of the plasmid vector COVID-eVax was carried out at Biomay (Austria) under current principles of good manufacturing practice (GMP) and following the guidelines for the “Manufacture of biologically active substances and medicinal products for human use” (Annex 2 of the EU-GMP guideline) and “Manufacture of investigational medicinal products” (Annex 13 of the EU-GMP guideline). The final concentration was adjusted to 4 ± 0.4 g/L product. The DNA was stored in 2 mL glass tubes at −20 °C and used for all experiments. The DNA was stored in PBS, pH of 7.3, at −20 °C, and tested for stability for 2 years.

Freeze-drying. The process was carried out using a VirTis SP Scientific Advantage EL-85 Freeze Dryer. Samples of COVID-eVax (500 μL) in 1.5 mL glass vials were allowed to freeze at −50 °C for 5 h under a light vacuum (about 500 mbar), and then the following conditions were applied:
1st step−50 °C150 mTorr2 h2nd step−40 °C100 mTorr2 h3rd step−30 °C100 mTorr2 h4th step−20 °C50 mTorr2 h5th step−10 °C50 mTorr2 h6th step−5 °C50 mTorr2 h7th step0 °C50 mTorr2 h8th step+10 °C50 mTorr2 h9th step+20 °C50 mTorr4 h

At the end of the sequence, the vacuum was broken by fluxing ultra-pure nitrogen, and vials were quickly capped under an inert atmosphere.

### 2.2. Analytical Methods

The molecular weight and the supercoiled/open coiled ratio of the plasmid DNA COVID-eVax were analyzed by running 200 ng of plasmid DNA on each lane on a 1% agarose TBE gel stained with SYBR-Safe DNA (Thermo Fisher Scientific, Carlsbad, CA, USA) for 2 h at 45 V. The gel images were acquired using a ChemiDoc^TM^ MP Imaging System, and quantification of the pDNA isoform ratios was determined through densitometric analysis using the software ImageLab 6.0.1 (Biorad Laboratories Inc, Hercules, CA, USA).

To set up the HPLC analytical method, reference standards were generated by enzymatic restriction; particularly, OC DNA was generated with Nt.BbvCI (New England Biolabs, Ipswich, MA, USA), and LN DNA was generated with HindIII-HF (New England Biolabs, Ipswich, MA, USA). A CIMac pDNA 0.3 mL weak anion-exchange analytical column (Sartorius AG, Gottinga, Germany) was used with a Shimadzu Prominence HPLC system (Shimadzu Corporation, Kyoto, Japan) equipped with an SIL-20ACHT UFLC autosampler, a column heater set to 25 °C, and an SPD-20AV detector. Absorbance was monitored at 260 nm, and the Shimadzu LabSolutions software was used for peak integration to obtain area and area %. The equilibration buffer (A) was 200 mM Tris–HCl with pH 8.0, and the elution buffer (B) was 200 mM Tris–HCl, 1 M NaCl, with pH 8.0. All buffers were filtered through 0.2 µm filters before use. The flow rate was 1.0 mL/min throughout. Each sample was injected in a 10 µL volume containing 1 µg SC DNA, 0.2 µg LN DNA, or 0.2 µg OC DNA. A linear gradient was developed from 65% to 85% buffer B in 15 min.

Potency Assay. HEK293 cells were transfected using the same amount of each DNA sample (4 μg), according to the Lipofectamine 2000 manufacturer’s instructions (ThermoFisher Scientific, Carlsbad, CA, USA) with cells at 80% confluence. Then, 72 h later, supernatants were collected and evaluated by Sandwich Elisa assay. Briefly, Maxisorp 96-well plates (Nunc) were coated with 1 ug/mL of the anti-RBD antibody 5B7 (generated in Takis) in PBS1X overnight at 4 °C, and after a wash in PBS1X-Tween 0.05% (PBST), they were blocked with 3% BSA-PBST 1 h at RT in agitation. Scalar doses of each supernatant in 1% BSA-PBST were added, and purified RBD [22] was used as the reference standard control. After o/n incubation at 4 °C, plates were washed, and the SARS-CoV-2 spike antibody, Rabbit PAb, Antigen Affinity Purified (Sino biological Cat: 40150-T62-COV2 100) was added and incubated for 3 h at RT. Finally, plates were washed and incubated with the Goat Anti-Rabbit IgG (H + L)-HRP Conjugate (Biorad) diluted 1:2000 in BSA 1%/PBST, 1 h at RT, and developed by adding 50 ul/well of TMB (3, 3′, 5, 5′ tetramethylbenzidine) liquid substrate (Sigma-Aldrich, St. Louis, MI, USA). Absorbance was measured at 450 nm using the ELISA plate reader Tecan (TecanGroup ltd, Mannedorf, Swiss). Data were plotted and analyzed using GraphPad to quantify RBD concentration in the supernatant of transfected HEK293 cells.

Analysis of Immune Responses. The in vivo experimental procedures were all approved by the local animal ethics council and the ethical committee of the Italian Ministry of Health, authorization # 586/2019-PR. The mouse experiments were conducted on 6-week-old C57Bl/6 female mice (Envigo, Indianapolis, Indiana, USA), as described in the figure legends. Mice were vaccinated with 10 μg of plasmid DNA delivered by EP in the quadriceps, following a prime-boost vaccination regimen (Days 0 and 28). DNA-EP was performed using a Cliniporator device (Igea, Carpi, Italy) and using a needle electrode (electrode N-10-4B), and the following electrical conditions were used: low voltage amplitude of 40 V (corresponding to an electric field strength of 100 V/cm), with 4 pulses of 5 msec duration separated by 5 msec intervals. At different time points, antibody and cell-mediated immune responses were analyzed.

ELISA Assay. The ELISA assay was performed as previously reported [22]. Briefly, the plates were functionalized by coating the RBD-6xHis protein and were blocked with 3% BSA-0.05% Tween 20-PBS. Mouse sera were added at a dilution of 1/300 in 1% BSA-0.05% Tween 20-PBS and diluted 1:3 up to 1/218700, in duplicate. After incubation overnight at 4 °C, antibody levels were detected using a secondary anti-murine IgG conjugated with alkaline phosphatase and the substrate for alkaline phosphatase yellow (pNPP) liquid substrate system for ELISA (cat. P7998 Sigma). After 30 min of incubation, the absorbance at 405 nm was measured using an ELISA reader. IgG antibody titers against the RBD protein were evaluated at two-time points (Days 28 and 35). Endpoint titers were calculated by plotting the log10 OD against the log10 sample dilution. A regression analysis of the linear part of the curve allowed the calculation of the endpoint titer. An OD of 0.2 was used as a threshold.

ELISpot Assay The assay was performed on splenocytes collected from vaccinated and control mice, according to the manufacturer’s instructions (Mabtech, Nacka Strands, Sweden) as previously described [27]. Splenocytes were plated at 4 × 10^5^ and 2 × 10^5^ cells/well, in duplicate, and stimulated with a pool of RBD peptides (JPT Peptide Technologies GmbH, Berlin, Germany) at a final concentration of 1 μg/mL. The next day, plates were developed according to the kit manufacturer’s instructions. The results were measured as spot-forming cells (SFC) per million splenocytes counted with an automated ELISPOT reader (Aelvis ELIspot reader, A.EL.VIS Gmbh, Hannover, Germany).

Statistical analysis. Mann–Whitney was utilized where indicated. All analyses were performed in GraphPad Prism 8.0.2 (Dotmatics, Boston, MA, USA)

## 3. Results

### 3.1. COVID-eVax Profile Changes as a Function of Storage Temperature and Lyophilization

To establish a stability profile, the plasmid COVID-eVax was placed in stable storage at a standard temperature of −20 °C. The DNA was analyzed at 0, 6, 12, 18, and 24 months for gene expression and compared with the engineering run as a reference. A potency assay for the evaluation of the protein expressed by the plasmid was established and validated (see Materials and Methods). COVID-eVax sequence encodes for the RBD fused to the TPA leader sequence, which drives the secretion of the protein in the supernatant of transfected HEK293 cells. As shown in Figure 1A, RBD produced at each timepoint matched the stability criterion, which was a variation in expression less than two standard deviations.

Having established that COVID-eVax is stable at −20 °C for at least 2 years without losing its physicochemical and gene expression properties, we wondered what impact OC DNA could have on the immune responses induced by DNA-EP.

To this end, we adopted the strategy used for an accelerated stability test [26]. The COVID-eVax plasmid was incubated at different temperatures (23 °C, 37 °C, 45 °C, and 65 °C) for 20 days, and then DNA integrity was analyzed by gel electrophoresis (Figure 1B). As expected, the percentage of OC DNA increased as a function of temperature, depicted by the gel electrophoresis and confirmed by HPLC analysis (Figure 1C). More specifically, the SC form showed a linear decrease from 84% to 12% in the DNA kept at 45 °C, while the OC form, as expected, increased in the opposite direction from 12% to 82%. The percentage of linear DNA was stable at around 5% up to 45 °C but increased dramatically to 62% at 65 °C. Then, we investigated the in vitro biological function by using the potency assay previously described. As shown in Figure 1D, a reduced gene expression was observed only with the highly degraded DNA kept at 65 °C. These results suggest that plasmid DNA carrying an increased percentage of OC DNA, such as that observed for the sample incubated at 45 °C, maintains its expression levels and biological functions.

### 3.2. COVID-eVax Immune Response Is Poorly Impacted by Molecular Isoform Composition

To evaluate the impact of DNA integrity on the immune response induced by electroporation delivery, C57Bl/6 mice were vaccinated with differently treated forms of COVID-eVax DNA, as reported in Figure 2, following the vaccination protocol depicted in Figure 2A. One week after the boost, RBD-specific T-cell responses were evaluated in the spleen upon overnight stimulation with the peptide pool covering the entire RBD sequence of the spike protein. Significant immune responses were observed in 60% (three out of five) of mice vaccinated with DNA kept at 23 °C and in 80% (four out of five) of mice vaccinated with DNA kept at 45 °C (Figure 2B,C).

No statistical difference was observed with respect to immune responses induced by reference DNA (−20 °C). Superimposable results were observed with the antibody response measured before the boost vaccination or after one week. The average antibody titer against RBD of mice vaccinated with DNA kept at 23 °C and 45 °C was comparable with that of DNA stored in the reference condition (−20 °C).

To further correlate the in vitro and in vivo biological functions with the percentage of OC DNA, we generated nicked DNA using a different protocol. The DNA formulated in PBS was dried out and kept as a powder for two weeks at room temperature. Then, the DNA was resuspended in injectable water and analyzed on an agarose gel for integrity. The results showed that in this condition the SC, OC, and LN isoforms were 56%, 44%, and 0%, respectively (Figure 3).

The potency assay showed a loss of expression by 38%. However, the antibody titer induced in mice vaccinated with the same immunization schedule depicted in Figure 2 showed similar antibody titers as reference DNA (Figure 3C). Taken together, these results show that the OC DNA, up to 82%, did not affect either the in vitro expression or the induction of immune response in vivo.

## 4. Discussion

While the pandemic has accelerated the development of new vaccine technologies, it has also shown the importance of making these vaccines equitably accessible to the people who need them. One of the factors that complicate the worldwide distribution and administration of COVID-19 vaccines is their need for a cold chain for shipment and storage. Although this issue has been faced in the smallpox global vaccination campaign [28], it became more evident for the novel mRNA-based vaccines that currently require continuous storage at ultra-cold temperatures, which is not feasible in many parts of the world. The need for thermostable vaccines had already been recognized before the COVID-19 pandemic, for example, in a study by the Vaccine Innovation Prioritization Strategy (VIPS) [29], in which the ability to withstand heat exposure was identified as the most desired characteristic for vaccines used in outreach and campaign settings by experienced immunization staff. Lack of vaccine thermostability leads to limited or no access to vaccinations by people living in remote areas or low-resource settings. It also leads to wastage when vaccines must be discarded after they have been (potentially) exposed to heat or freezing, which is the main cost driver, adding to the costs of the cold chain itself. There might even be a risk that vaccines with reduced potency are administered, leaving people vulnerable to disease. Alternatively, to facilitate local vaccine production, BioNTech has planned to ship vaccine production units called “BioNTainers” to Africa and other regions left behind during the pandemic. However, this approach is costly and requires a complex implementation.

Unlike mRNA, DNA is considered to be thermostable and does not need the cold chain for transport and storage in environments and countries with limited resources, an essential feature that affects the overall cost of these treatments. The cost of cold chain for DNA stability is an issue that has also been addressed in the context of long-term information storage [26,30]. The research on analytical methods of DNA integrity in different storage conditions has produced a series of methods to evaluate DNA integrity. The same methods are used in gene therapy and genetic vaccination approaches. Pharmacopeia requires DNA integrity above 80%, defined as the percentage of supercoiled plasmid DNA.

In this study, we show that the plasmid DNA coronavirus vaccine COVID-eVax delivered by EP can be stored for 20 days at room temperature, ranging from 23 °C to 45 °C, maintaining biological functions in vitro such as gene expression and in vivo, as well as induced immune responses. The accelerated stability protocol produced the expected increase in DNA nicking (Figure 1). The DNA kept for 20 days at 45 °C showed similar efficacy results in terms of T cells and antibody responses to those of the GMP-like reference DNA (Figure 2). Although the number of mice responding to the vaccination was reduced, the level of the immune responses was not affected, suggesting that it is more a technical variability. In line with this, the experiment with lyophilized DNA showed a more uniform result (Figure 3). Indeed, EP technology is prone to a certain level of variability, as we have observed in murine cancer models [31]. Moreover, it is worth mentioning the indirect evidence observed with linear DNA produced by PCR, which is as immunogenic and efficacious as supercoiled plasmid DNA [18]. In this recent paper, we compared side-by-side PCR-produced linear DNA and plasmid DNA in gene expression and a cancer vaccine model and showed that the two forms of DNA vaccine delivered by EP resulted in similar immune responses and antitumor effects.

The study was conducted using a low dose of COVID-eVax (10 μg) in line with the doses that have been used in preclinical studies [22] and then in the human clinical trial [24]. The impact of EP is more appreciable at this low DNA dose rather than at saturating dose where there is almost no difference between DNA alone and DNA-EP [32]. Immunity is driven by EP-induced inflammation that is more evident at low doses [33] rather than at high DNA doses where the most of inflammation is induced by the DNA itself [34]. We cannot exclude that other factors such as different plasmid composition, electroporation protocols, site of muscle injections, and assay methods among other differences may affect the efficacy of OC DNA delivered by EP. Additional experiments are required to clarify all these relevant technical points.

Our results also support the possibility of using COVID-eVax stored at −20 °C for a long time as suggested by the results of DNA kept at 45 °C, which corresponds to storage for 5 years at −20 °C [26]. The DNA-EP delivery system is currently used in many clinical trials for which a stability test is mandatory. Our data investigated for the first time the impact of OC DNA on in vitro expression as a potency assay and the immune responses induced by DNA-EP in a mouse model. Our results suggest that the stringent requirements for DNA stability can be revisited, at least for this delivery system. DNA-EP is an efficient and safe technique allowing the delivery of drugs or genetic material into target cells. The possibility of using DNA vaccines without an expensive cold chain may have an economic impact on countries with limited resources for the distribution of pandemic vaccines. Reduced logistic requirements may enlarge the distribution to remote populations and, therefore, may have a very positive impact at the population level.

## Figures and Tables

**Figure 1 vaccines-11-00678-f001:**
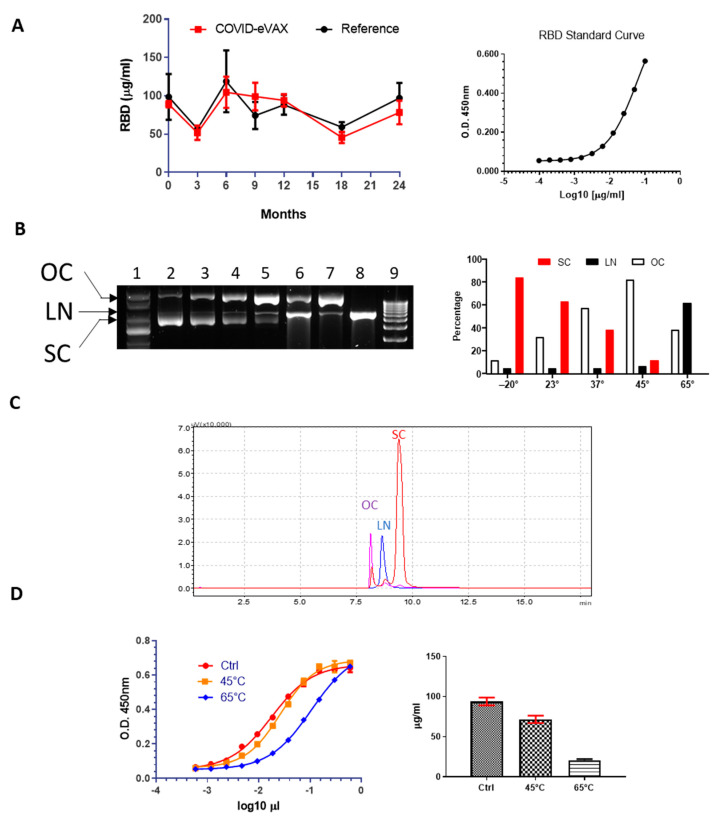
In vitro characterization of accelerated DNA stability: (**A**) Potency assay with GMP COVIDeVax with respect to reference engineering run DNA (reference) overtime and standard curve with purified RBD; (**B**) Agarose gel electrophoresis of the different COVIDeVax forms stored at different temperatures: (1) supercoiled marker DNA, (2) DNA stored at −20 °C, (3) DNA kept at 23 °C, (4) DNA kept at 37 °C, (5) DNA kept at 45 °C, (6) DNA kept at 65 °C, (7) control nicked DNA, (8) control linearized DNA, (9) 1 kb linear DNA ladder; on the right panel of the figure, quantification of the different DNA forms at selected temperatures is shown; (**C**) Representative overlayed chromatograms of OC, LN and SC forms of COVID-eVax; (**D**) Potency assay of two representative temperatures (45 °C and 65 °C), standard curve (Ctrl), and quantification are shown in the graph.

**Figure 2 vaccines-11-00678-f002:**
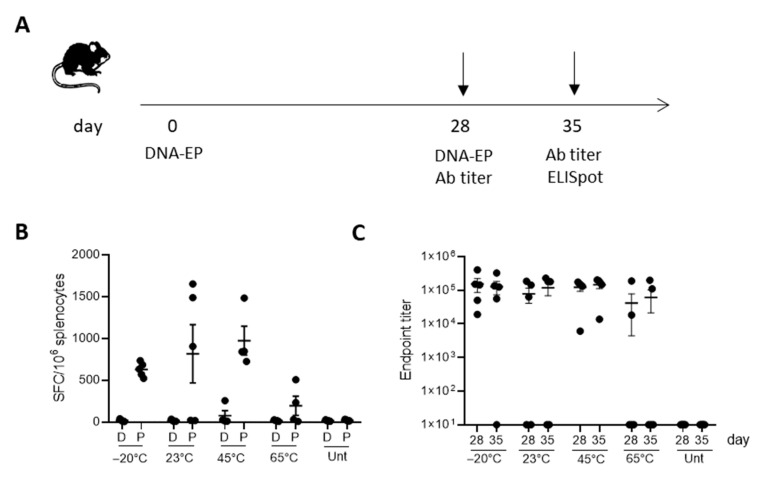
Immunogenicity of OC COVID-eVax vaccine delivered by EP: (**A**) Groups of C57Bl/6 mice (*n* = 5) were vaccinated on Days 0 and 28 and sacrificed on Day 35 for IFN-γ ELISpot analysis. Sera were collected on Days 28 and 35 for binding antibodies titer analysis: (**B**) IFNγ-producing T cells measured by ELISpot assay on splenocytes restimulated with RBD peptide pool; (**C**) Total RBD-specific IgG endpoint titers measured by ELISA assay in sera collected on Days 28 and 35. Each symbol represents an individual sample with the error bars representing the SEM. Significance was determined using Mann–Whitney test.

**Figure 3 vaccines-11-00678-f003:**
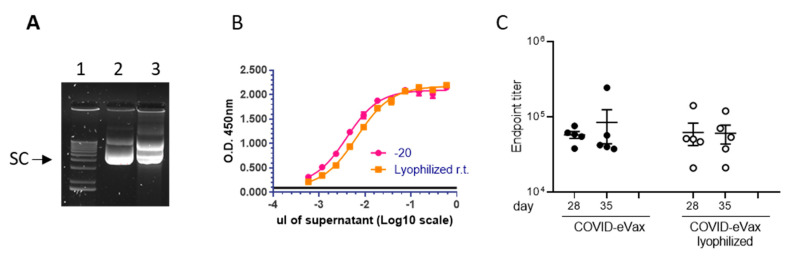
Lyophilization process impact on in vitro and in vivo biological functions: (**A**) DNA was lyophilized to induce DNA nicking (Materials and Methods): (1) supercoiled marker DNA, (2) DNA stored at −20 °C, (3) DNA lyophilized; (**B**) Potency assay showing the gene expression of DNA in panel A; (**C**) Endpoint titers measured by ELISA assay of C57Bl/6 mice (*n* = 5) vaccinated via the DNA-EP depicted in Figure 1. Each symbol represents an individual sample with the error bars representing the SEM.

## Data Availability

There are no additional data.

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
