# Peer review of "Immunogenicity of COVID-eVax Delivered by Electroporation Is Moderately Impacted by Temperature and Molecular Isoforms"

_vaccines, 2023, doi:10.3390/vaccines11030678_

Round 1
Reviewer 1 Report (Previous Reviewer 1)
To Authors:
The manuscript “Immunogenicity of COVID-eVax delivered by electroporation is moderately impacted by temperature and molecular isoforms” by D’alessio F. and co-authors is the resubmission of the revised version of manuscript “Vaccines-2026698”.
The authors did a very good job correcting the style of the writing, adding an informative introduction and discussion , and editing other issues.
However, there are a few major questions and some minor corrections that need to be addressed.
Main questions and issues:
1. The description of incubation conditions for the stability test needs to be added in methods because Figure 1B shows the temperature as 45o but Figure 1D shows 42 o. Please correct it also in the figure legends and text.
2. Authors need to describe in methods the way how the results on the graph Figure 1B were obtained from the image of the bands.
3. Authors need to explain why the scale of the protein expression in Figure 3B is different and way higher than the standard curve (Figure 1A) and test o Figure 1D. It is important to show that it is still in a linear zone and explain the difference and potential effect on the outcome of vaccination.
Minor issues that need editing:
4. There are still a few phrases that need additional editing: some examples are lines 35-36; line 42; lines 51-52; lines 57-58, line 158, line 207, and more. I would suggest another round of proof-reading of this version of the manuscript
5. In the “Material and Methods” part about “Potency Assay”, starting at line116, missing information about culturing and plating conditions for Hek cells which is very critical for understanding this specific method.
6. Line 140-141 is missing catalog numbers for the kits that were used in the assay.
7. Authors need to describe in methods the way how the results on the graph Figure 1B were obtained from the image of the bands.
8. The description of incubation conditions need to be described in methods because Figure 1B show the temperature was 45o but Figure 1D shows 42 o. Please correct it also in the figure legends and text. To avoid confusion it can be useful to describe tested conditions in the “Materials and methods”.
9. Please, rephrase the title of section 3.2. (line 188)
10. Please, rephrase “titer of mice…” line 209.
11. Missing the labeling of the gel’s columns in Figure 3A
Author Response
Main questions and issues:
- The description of incubation conditions for stability test needs to be added in methods because Figures 1B show temperature was 45o but Figure 1D shows 42 o. Please correct it also in the figure legends and text.
We thank the reviewer for noticing this discrepancy the figure and figure legend was corrected and text was added to the result (lane 174).
2 Authors need to describe in methods the way how the results on the graph Figure 1B were obtained from the image of the bands.
This information is reported in Material and Methods (lane 102-104) The gel images were acquired using a ChemiDocTM MP Imaging System and quantification of the pDNA isoform ratios was determined through densitometric analysis using the software ImageLab 6.0.1 (BioRad, USA)
We added in Material and Methods the software used for statistical analysis and graphs (lane 152-153)
- Authors need to explain why the scale of the protein expression on the Figure 3B is different and way higher than the standard curve (Figure 1A) and test o Figure 1D. It is important to show that it is still in linear zone and explain the difference and potential effect on the outcome of vaccination.
The reviewer noticed an apparent discrepancy, which is explained as follows. The different OD is due to the reaction conditions that have been established according to the regulatory requirements We used this test for the release of DNA batch for the human clinical trial. To match the stringent criteria of minimal variation with respect to the engineering run (less than two standard deviations, see lane 161-162) the assay is carried out at a fixed time of 1 h. at r.t. (lane 124). Since the readout is an enzymatic reaction, the OD can be different, as noticed by the reviewer, but the sigmoid curve was determined allowing the calculation of protein concentration.
Minor issues that need editing:
- There are still few phrases that need additional editing: some of
examples lines 35-36; line 42; lines 51-52; lines 57-58, line 158 , line
207 and more. I would suggest another round of proof-reading of this version of the manuscript.
New proofreading was performed by a mother language assistant.
- In the “Material and Methods” part about “Potency Assay”, starting
at line116, missing information about culturing and plating conditions for Hek cells which is very critical for understanding this specific method.
The condition of cell plating was specified (lane 117)
- Line 140-141 is missing catalog numbers for the kits that were used
in assay.
Kit number added
- Authors need to describe in methods the way how the results on the graph Figure 1B were obtained from the image of the bands.
This information is reported in Materials and Methods (lane 102-104) The gel images were acquired using the ChemiDocTM MP Imaging System and quantification of the pDNA isoform ratios was determined through densitometric analysis using the software ImageLab 6.0.1 (BioRad, USA)
We added in Material and Methods the software used for statistical analysis and graphs (lane 152-153)
- The description of incubation conditions need to be described in methods because Figures 1B show temperature was 45o but Figure 1D shows
42 o. Please correct it also in the figure legends and text. To avoid confusion it can be useful to described tested conditions in the “Materials and methods”.
We thank the reviewer for noticing this discrepancy the figure and figure legend was corrected and text was added to the result (lane 174).
- Re-phrase the title of the section 3.2. (line 188)
We respectfully disagree with the reviewer, the title is essential to the paper's message, and we consider it appropriate.
- Please rephrase “titer of mice…” line 209.
The phrase was modified as follows: ”The average antibody titer against RBD of mice vaccinated with
- The labeling of the gel’s columns in Figure 3A
Lanes were numbered and descriptions were added to the legend
Reviewer 2 Report (New Reviewer)
The present work entitled: “Immunogenicity of COVID-eVax Delivered by Electroporation Is Moderately Impacted by Temperature and Molecular Isoforms” by Federico D’Alessio and his team describes the impact of OC DNA on in vitro expression as a potency assay and the immune responses induced by DNA-EP in a mouse model. The study highlighted Immunogenicity of COVID-eVax and core issues in vaccine delivery and efficacy against COVID.
The work in the manuscript is correctly performed, the Introduction is appropriate and well summarizes the topic.
The employed methodologies are accurate and correspond to the purpose of the experiments.
The results are correctly implemented as per the methodology used in the said parameters.
Discussion sufficiently covers the justification of the proposed study.
References are sufficiently updated.
I recommend this Manuscript for publication in MDPI (Vaccines) after moderate English editing and proofreading.
Author Response
New proofreading was performed by a mother language assistant.
Reviewer 3 Report (New Reviewer)
The authors investigate if the status of the plasmid COVID-eVax has an influence on the quality of the vaccine when administered in mice.
Using readouts such as Elispot assay or antigen-specific antibody titres, the authors find that the immune response to the DNA-coded antigen is only moderately impacted by temperature and molecular isoforms.
Lyophilisation or heat treatment can convert supercoiled (SC) plasmid to open circular (OC) plasmid DNA (single break in the DNA backbone) or to linear (LN) DNA (double nick on both strand). Interestingly, immune responses in vivo are not dramatically influenced although in an in vitro potency assay antigen levels produce after plasmid transfections are lower. This is an important observation for the availability and accessibility of a functional DNA vaccine.
Concerns:
1) The authors should state in Fig. 1 D for which plasmid concentration the antigen production was quantified.
2) Fig. 2. The data on reactive T-cells and antibody production do not always correlate (experimental data for -20°C and 45°C). The authors should comment on this partial discrepancy.
3) Fig. 3: "The potency assay showed a loss of expression by 38%" when lyophilized DNA was used. The authors should state in Fig. 2B for which plasmid concentration antigen production was quantified.
4) Discussion: "Unlike mRNA, DNA is considered to be thermostable and does not need the cold chain for transport and storage in environments and countries with limited resources, an essential feature that affects the overall cost of these treatments."
I think it is quite accepted that mRNA is very stable (if there are no RNases present), even more stable than DNA. I would suggest that the authors soften their claim here.
Author Response
1) The authors should state in Fig. 1 D for which plasmid concentration the antigen production was
quantified.
This information was added in M&M (lane 132)
2) Fig. 2. The data on reactive T-cells and antibody production do not always correlate (experimental data
for -20°C and 45°C). The authors should comment on this partial discrepancy.
Thank you for noticing this aspect, to address this point the following phrase was added: The marginal
differences between antibody responses and T cell responses are likely due to the different sensitivity of the
ELISA versus the ELIspot assay.
3) Fig. 3: "The potency assay showed a loss of expression by 38%" when lyophilized DNA was used. The
authors should state in Fig. 2B for which plasmid concentration antigen production was quantified.
This information was added in M&M (lane 132)
4) Discussion: "Unlike mRNA, DNA is considered to be thermostable and does not need the cold chain for
transport and storage in environments and countries with limited resources, an essential feature that
affects the overall cost of these treatments."
I think it is quite accepted that mRNA is very stable (if there are no RNases present), even more stable than
DNA. I would suggest that the authors soften their claim here.
We agree with the referee, purified mRNA is stable and may have a similar level of conservation as pDNA. We
meant a comparison between nacked DNA used in the delivery system with electroporation and formulated
RNA, the phrase was modified accordingly: “Genetic vaccines based on nacked DNA are considered to be
thermostable and do not need the ultra-cold chain for transport and storage as is the case for formulated
mRNA or vectored vaccines, a feature that affects the overall cost of these treatments in countries with
limited resources.”
This manuscript is a resubmission of an earlier submission. The following is a list of the peer review reports and author responses from that submission.
Round 1
Reviewer 1 Report
In the manuscript “Immunogenicity of COVID-eVax is moderately impacted by 2 temperature and molecular isoforms” D’alessio F. and co-authors are evaluating the relationship between the accumulation of nicked and linear DNA of DNA vaccine (COVID-eVAx) using different temperatures for storage on the immunogenic property.
The authors have a very interesting idea to address the question of the stability of DNA-vaccine using methods described for the studies of DNA for data storage. This manuscript needs a very good review of English grammar and scientific style. Only a few examples: phrase 166 165 (“the SC form showed a linear decrease..”); phrase 167-168 (“aa reduced gene expression…”
There is a substantial absence of correct descriptions and legends for all figures: for instance in Figure 1 – a few graphs have no labeling and description at all and, it is even hard to reference it, “the second part of part A”. The same figure 1 has no correct labeling for figure B and of the exes for figure C.
The introduction and discussions both lack a good review of any work done to determine the stability and immunogenicity of DNA plasmids and DNA vaccines as well as a description of the best practice methods to determine the stability of vaccines. There is only one reference to some review-type of work. There is also a lack of references for the methods used for tests like ELIZA, ELISpot, “Analysis of immune response” etc. Some methods and results are lacking descriptions of data analysis:
Phrase 132- 133 how was the titer calculated?
Figure 2a- the description (182-183) does not correlate with the graph. What does it mean:”data are from one out of two experiments (184-185)”?? Which of the experiments, how it was selected, and how many samples were used? The reader has to guess it or try to find it in other parts of the manuscript.
Also, I am not sure that the authors addressed correctly the question about the ability of damaged DNA to produce an immunological response. There are no clear controls showing the sensitivity of those tests for only open-circled DNA and linear DNA. It is not clear what the minimal amount of SS_ DNA is needed to produce a correct response in mice using this method of vaccination.
This manuscript needs more discussion about the significance of values in Figures 2A and B. The conclusion “Although the number of mice responding to vaccine… (235-236) needs to be explained with references and examples. It very important to confirm that the results are correct because it is well known that mice are not very good models for the immunogenicity of the genetic material.
Author Response
Reviewer 1
In the manuscript “Immunogenicity of COVID-eVax is moderately impacted by 2 temperature and molecular isoforms” D’alessio F. and co-authors are evaluating the relationship between the accumulation of nicked and linear DNA of DNA vaccine (COVID-eVAx) using different temperatures for storage on the immunogenic property.
The authors have a very interesting idea to address the question of the stability of DNA-vaccine using methods described for the studies of DNA for data storage. This manuscript needs a very good review of English grammar and scientific style. Only a few examples: phrase 166 165 (“the SC form showed a linear decrease..”); phrase 167-168 (“aa reduced gene expression…”
Thanks for the note. The reviewer is likely referring to the phrase in lane 157 “the SC form showed a linear decrease” and not 165-166, similarly, the phrase “A reduced gene expression” in 160-161 and not 167-168. However, we sent the manuscript to professional English proofreading and has been accordingly revised, as suggested.
There is a substantial absence of correct descriptions and legends for all figures: for instance in Figure 1 – a few graphs have no labeling and description at all and, it is even hard to reference it, “the second part of part A”. The same figure 1 has no correct labeling for figure B and of the exes for figure C.
We thank the reviewer for the comments. We have made the following corrections in Fig. 1
- The HPLC chromatogram was updated
- The lanes of the agarose gel were correctly assigned, and the legend of the figure was modified accordingly
The introduction and discussions both lack a good review of any work done to determine the stability and immunogenicity of DNA plasmids and DNA vaccines as well as a description of the best practice methods to determine the stability of vaccines. There is only one reference to some review-type of work. There is also a lack of references for the methods used for tests like ELIZA, ELISpot, “Analysis of immune response” etc. Some methods and results are lacking descriptions of data analysis:
The comment is well taken. For the issue of DNA stability and storage of information, we have added the reference of Matange et al 2021 in the discussion. For the ELISA and ELISPOT we have cited our published paper where the method is described more in detail (Conforti et al 2021 and Lione et al 2021, respectively).
Phrase 132- 133 how was the titer calculated?
The calculation method was described in M&M lane 125-127 “Endpoint titers were calculated by plotting the log10 OD against log10 sample dilution. A regression analysis of the linear part of the curve allowed the calculation of the endpoint titer. An OD of 0.2 was used as a threshold.”
Figure 2a- the description (182-183) does not correlate with the graph. What does it mean:”data are from one out of two experiments (184-185)”?? Which of the experiments, how it was selected, and how many samples were used? The reader has to guess it or try to find it in other parts of the manuscript.
To answer this comment, we have specified that “both data” meaning T cell and antibody were from one experiment. As indicated in the figure legend the number of mice is described as "(n=5)” and the individual values as “individual samples”. We have canceled “DMSO (D) or”, which is left over from a previous version.
Also, I am not sure that the authors addressed correctly the question about the ability of damaged DNA to produce an immunological response. There are no clear controls showing the sensitivity of those tests for only open-circled DNA and linear DNA. It is not clear what the minimal amount of SS_ DNA is needed to produce a correct response in mice using this method of vaccination.
We measured immunogenicity as a function of different percentages of nicked DNA induced in an accelerated stability test that is accepted as the release test of injectable DNA drugs. To address the point raised by the reviewer we have cited in the discussion our recent work on linear DNA delivered by electroporation in the field of personalized cancer vaccines. In the paper by Conforti et al, we clearly show that linear DNA is as efficient as plasmid DNA in the electroporation delivery protocol. The results with nicked DNA produced by both accelerated stability protocol and dried DNA are consistent with the immune responses induced by linear DNA (Conforti et al J Exp Clin Cancer Res. 2022).
This manuscript needs more discussion about the significance of values in Figures 2A and B. The conclusion “Although the number of mice responding to vaccine… (235-236) needs to be explained with references and examples. It very important to confirm that the results are correct because it is well known that mice are not very good models for the immunogenicity of the genetic material.
The technical variability was observed also in other papers, we cited the data by Salvatori et al 2022.
Reviewer 2 Report
This study verified the long-term stability of COVID-eVax, a plasmid DNA vaccine. It is commendable in that immunogenicity can be maintained even when heat treatment is applied. The following points should be clarified.
1 Electroporation was used in this study to introduce the plasmid DNA vaccine into animals. However, if there are results of verification of the effect when the vaccine is injected intramuscularly, assuming clinical trials, show its detail.
2 Regarding the issue of the heat resistance of vaccines, it is stated that the COVID-19 vaccine is the first vaccine. However, a similar issue was raised in the WHO's smallpox eradication program, and heat resistance tests are required for smallpox vaccines. Therefore, this issue should be described as an issue for pandemic vaccines and not limited to COVID-19.
Breman JG. Smallpox J Inf Dis. 2021; 224:S379–86. https://doi.org/10.1093/infdis/jiaa588
Author Response
Reviewer 2
This study verified the long-term stability of COVID-eVax, a plasmid DNA vaccine. It is commendable in that immunogenicity can be maintained even when heat treatment is applied. The following points should be clarified.
1 Electroporation was used in this study to introduce the plasmid DNA vaccine into animals. However, if there are results of verification of the effect when the vaccine is injected intramuscularly, assuming clinical trials, show its detail.
The clinical data have been submitted for publication and are currently under the embargo of the Journal. We expect the revised version to be accepted for publication this month. Data are encouraging and show induction of antibodies and T cell immune response in 68 volunteers.
2 Regarding the issue of the heat resistance of vaccines, it is stated that the COVID-19 vaccine is the first vaccine. However, a similar issue was raised in the WHO's smallpox eradication program, and heat resistance tests are required for smallpox vaccines. Therefore, this issue should be described as an issue for pandemic vaccines and not limited to COVID-19.
We thank the reviewer for this helpful suggestion: the smallpox study was mentioned in the discussion suggesting that cold chain is a general issue in facing pandemic infections